# MediBench: A Benchmark for VAEs in Medical Imaging Across Fidelity, Structure, and Latent Utility

## Abstract

High-resolution medical images pose considerable computational challenges for training deep learning models. While modern architectures continue to achieve strong performance, these demands have motivated a shift toward latent space–based approaches. Particularly in generative modeling, Variational Autoencoders (VAEs) provide an efficient foundation for representation learning. The effectiveness of this entire paradigm, however, is contingent upon the VAE's ability to fulfill a *dual mandate*: preserving sufficient information for downstream understanding tasks while enabling high-fidelity image generation. Despite the central role of this dual capability, the medical imaging community lacks a standardized framework for its systematic evaluation. To fill this gap, we introduce **MediBench**, a comprehensive benchmark designed to systematically evaluate how existing VAEs perform in the medical domain. Our framework evaluates VAEs across three pillars: (1) Reconstruction Fidelity and (2) Clinical Structure Preservation to evaluate whether reconstructions maintain essential clinically relevant structures, and (3) Latent Space Utility to measure the effectiveness of the learned latent space in supporting clinically relevant downstream analyses. We conduct an extensive evaluation on a diverse suite of medical datasets, comparing a wide range of general-purpose and medical-specific VAE architectures across 2D and 3D modalities. Our analysis reveals consistent trade-offs across the three pillars. Tokenized and vector quantized VAEs learn stronger latents than continuous VAEs. Medical pretraining improves transfer and structural preservation. Higher pixel fidelity often does not translate into downstream gains. MediBench provides a standardized and clinically grounded tool for selecting and developing VAEs in medicine. It advances reliable and efficient foundation models for medical AI.

## 1 Introduction

The resolution of medical imaging keeps rising. Digital pathology reaches gigapixel scales. Radiology acquires full volumes in CT and MRI. These advances give clinicians unprecedented anatomical and pathological detail (Patil et al., 2024). They also introduce heavy computational costs that slow down model development and deployment (Sarki et al., 2023). As a result, many pipelines move computation to compact latent spaces instead of raw pixels. This shift powers both *generative* tasks, such as high-fidelity synthesis with latent diffusion (Rombach et al., 2022), and *understanding* tasks, such as disease classification and segmentation (Marisca et al., 2023). Variational Autoencoders (VAEs) sit at the center of this paradigm. They produce efficient and informative latent representations. The quality of these latents largely determines the reliability of later generative and discriminative models.

The role of the VAE introduces a rate versus distortion trade-off. The encoder must retain task-relevant information while keeping reconstruction errors low. Improving one objective does not guarantee the other. Aggressive compression may blur thin vessels or under-encode low-contrast lesions. In practice, many medical pipelines reuse VAEs trained on natural images (StabilityAI, 2022). This creates cross-domain shifts in intensity calibration, resolution, and protocol. Small or rare findings can be affected disproportionately. Downstream linear probes and segmentation

Figure 1: **Benchmarking data.** Modalities and downstream tasks included in MediBench. We cover histopathology, dermatoscopy, fundus photography, chest X-ray, and 3D MRI/CT cohorts with clinically meaningful tasks (e.g., tumor detection, disease grading, anatomical segmentation).

can also change (Guan & Liu, 2021). These issues motivate a principled evaluation that measures reconstruction fidelity, clinical structure preservation, and latent utility.

From a practical standpoint, VAEs are now central to medical AI. Pathology scanners produce gigapixel slides. Radiology systems handle thousands of 3D scans per day. Direct pixel-space modeling is often infeasible. Compact latents reduce storage and transmission. They speed up training and inference. They also enable real-time or resource-constrained deployment. If a VAE fails to preserve critical structures, the downstream pipeline suffers. Diagnosis and planning can be compromised. A medical VAE benchmark is therefore useful in practice. It should tell practitioners which models balance compactness and clinical fidelity.

Scientifically, systematic evaluation of medical VAEs remains limited. Model designs are diverse. Reports differ in datasets, resolutions, and training regimes. Results are hard to compare across studies. This echoes computer vision before ImageNet. Progress existed, but it was difficult to measure. A standardized medical VAE benchmark can change this. It can align evaluation across modalities and tasks. It can turn isolated improvements into cumulative progress.

Medical imaging also poses unique challenges. Downstream tasks are sensitive to subtle structural errors. A blurred branch or a missed small lesion may change interpretation. Compression levels that look harmless on natural images may erase diagnostic patterns. Natural-image benchmarks do not capture these risks. A medical benchmark should therefore be judged by practical utility. It should help choose the right VAE for clinical or research use. It should balance efficiency and reliability.

Common metrics have limits. PSNR and FID are convenient, but they measure pixels or distributions (Zhang et al., 2018). They can miss structural errors that matter clinically. They also say nothing about how useful the latent is for classification or segmentation. This gap calls for a framework that evaluates complementary dimensions tied to clinical plausibility and task performance.

We introduce **MediBench** to meet this need. The benchmark evaluates VAEs against their dual mandate in medicine. It covers diverse modalities and resolutions. It separates competing goals and reports clear summaries. First, we assess **reconstruction fidelity** with SSIM and PSNR on held-out data using consistent normalization and subject-wise splits. This yields a concise view of rate vs. distortion under clinically relevant resolutions. Second, we evaluate **clinical structure preservation**. We generate label maps on original and reconstructed images using robust pretrained segmenters. We then compare structure-focused metrics, including model-provided quality scores and label agreement. This targets semantically meaningful anatomy and aligns with clinical reading. Third, we measure **latent representation utility**. We train lightweight classifiers on VAE latents and compare to image-based baselines on raw inputs. For 2D we report Accuracy, Macro F1, Precision, Recall, and Macro AUC. For 3D we compare Image Accuracy and Latent Accuracy under standardized protocols. This isolates the value of the representation and highlights when compact latents remain discriminative.

With MediBench, we run a head to head comparison of medical specific and general purpose VAEs in both 2D and 3D. We cover multiple modalities and report both per dataset and aggregated results. The benchmark exposes clear patterns: tokenized and vector quantized models deliver stronger latent utility, medical pretraining improves transfer, and gains in SSIM or PSNR do not guarantee downstream performance. Structure aware metrics capture failure modes that pixel metrics miss. MediBench therefore enables principled model selection and development for medical AI.

## 2 RELATED WORK

### 2.1 VAEs AND LATENT SPACE MODELING IN MEDICAL IMAGING

Variational Autoencoders (VAEs) have been central to modern generative modeling and representation learning since their introduction by Kingma & Welling (2014). By optimizing the ELBO to balance reconstruction and latent regularization, VAEs learn structured latent spaces that support both synthesis and analysis. Beyond the original continuous formulation, a rich line of discrete/quantized models replaces Gaussian latents with codebooks. VQ-VAE and VQ-VAE-2 alleviate posterior collapse and yield sharper reconstructions through a categorical bottleneck (Van Den Oord et al., 2017; Razavi et al., 2019), while VQ-GAN adds perceptual and adversarial terms to recover high-frequency detail without sacrificing a learnable latent interface (Esser et al., 2021). Residual or multi-stage quantization increases effective capacity by composing quantizers (Lee et al., 2022). Recent work further casts quantization itself as stochastic variational inference: SQ-VAE improves codebook utilization and stability without heuristic resets (Takida et al., 2022) and explores tokenization without explicit codebooks via binary/spherical constraints for extremely compact visual tokens (Zhao et al., 2024).

Progress on priors and inference complements these designs. Hierarchical and multi-scale decoders (NVAE/VDVAE-style pyramids) factorize latents across depth to capture global-to-local structure; flow-augmented inference and flow/autoregressive priors shrink amortization gaps and better fit complex posteriors; PixelVAE-style decoders model high-frequency detail while latents encode semantics. Objective-side refinements such as $\beta$-VAE/FactorVAE for disentanglement and WAE/InfoVAE-style distribution matching, together with practical recipes like KL annealing, free bits, and perceptual losses, mitigate blurriness or collapse and improve downstream utility. Two recent peer-reviewed surveys synthesize these axes: Foo et al. (2023) situates hierarchical, discrete/quantized, and hybrid VAE families within contemporary AIGC pipelines, and Teli (2025) emphasizes clinically motivated variants that promote sparse/structured latents, sharpen reconstructions with adversarial or perceptual terms, and integrate heterogeneous inputs.

These properties are especially attractive in medical imaging, where data are high dimensional and labels expensive. On the generative side, VAEs and hybrids synthesize data to balance rare cohorts and reconstruct missing modalities, from PET-from-MRI translation to diffusion-augmented pipelines operating in compact latents (Pan et al., 2018; Meng et al., 2024). On the analytical side, VAE-derived features support classification, anomaly detection, and interpretable diagnosis; examples include attribute-aligned latents for myocardial infarction prediction and reconstruction-error screening for head-CT quality control (Marisca et al., 2023; Ghosh et al., 2023). Together, these results underscore the dual role of VAEs in medicine: enabling realistic, controllable image generation while providing compact, discriminative representations for downstream analysis.

### 2.2 BENCHMARKING VAEs AND VISUAL REPRESENTATIONS

Standardized evaluation of VAEs has lagged behind their adoption. In the general domain, TokBench examines whether VAEs preserve fine-grained cues such as text and faces (Wu et al., 2025), and VBench organizes video evaluation along perceptual, motion, and fidelity dimensions (Wang et al., 2024). These protocols, however, do not directly address medical idiosyncrasies: domain shift across scanners and protocols, limited labels, and strict requirements for robustness and interpretability (Ghassemi et al., 2020). Moreover, pixel/distribution scores like PSNR or FID may miss subtle but clinically consequential structural errors and say little about latent informativeness for analysis (Zhang et al., 2018).

Motivated by these gaps, our benchmark adopts three complementary pillars—reconstruction fidelity, clinical structure preservation, and latent-space utility—under consistent, modality-aware protocols. Full definitions and implementation details appear in Sections 3 and 4.

## 3 MEDIBENCH: FRAMEWORK AND METRICS

To systematically assess the fitness of Variational Autoencoders (VAEs) for their dual mandate in medicine, we introduce **MediBench**, a comprehensive evaluation framework. This section lays out the methodological foundation of our benchmark. We begin by providing the necessary technical preliminaries on VAE architectures, then detail our unified evaluation pipeline, and finally, provide formal definitions and computational details for each of our evaluation metrics.

### 3.1 PRELIMINARIES: VAES AND VQ-VAES

Our benchmark evaluates two primary families of autoencoder architectures that are foundational to modern latent space-based models.

**Variational Autoencoders (VAEs).** The standard VAE (Kingma & Welling, 2014) learns a mapping to a continuous latent space. It consists of a probabilistic encoder $q_\phi(\mathbf{z}|\mathbf{x})$ that models the distribution of the latent vector $\mathbf{z}$ given an input image $\mathbf{x}$, and a probabilistic decoder $p_\theta(\mathbf{x}|\mathbf{z})$ that models the distribution of the image given the latent vector. The encoder typically outputs the parameters of a diagonal Gaussian distribution, $\mathcal{N}(\boldsymbol{\mu}, \boldsymbol{\sigma}^2\mathbf{I})$. The model is trained by maximizing the Evidence Lower Bound (ELBO) on the data log-likelihood:

$$\mathcal{L}_{\text{ELBO}}(\theta, \phi; \mathbf{x}) = \mathbb{E}_{q_\phi(\mathbf{z}|\mathbf{x})}[\log p_\theta(\mathbf{x}|\mathbf{z})] - D_{\text{KL}}(q_\phi(\mathbf{z}|\mathbf{x})||p(\mathbf{z})) \tag{1}$$

where the first term is the expected reconstruction log-likelihood and the second is the Kullback-Leibler (KL) divergence between the encoder's output distribution and a prior $p(\mathbf{z})$, usually a standard normal distribution.

**Vector-Quantised Variational Autoencoders (VQ-VAEs).** In contrast, the VQ-VAE (Van Den Oord et al., 2017) learns a discrete latent representation. Its encoder $E(\mathbf{x})$ produces a continuous output feature map $\mathbf{z}_e(x) \in \mathbb{R}^{h \times w \times d}$. This output is then quantized by replacing each vector with its nearest neighbor from a learned, finite codebook of embeddings $\mathcal{E} = \{e_k\}_{k=1}^K, e_k \in \mathbb{R}^d$. The quantization process for a vector $z_{ij}$ at a spatial location $(i, j)$ can be written as:

$$\mathbf{z}_q(x)_{ij} = \text{Quantize}(z_e(x)_{ij}) = e_k \quad \text{where} \quad k = \arg\min_j ||z_e(x)_{ij} - e_j||_2 \tag{2}$$

The decoder then reconstructs the image from this quantized feature map $\mathbf{z}_q(x)$. This discrete representation is often better at preserving sharp details in reconstructions. Both architectures are assessed on their ability to perform Image Reconstruction and Feature Encoding.

### 3.2 EVALUATION PIPELINE

The core of MediBench is a unified pipeline that assesses a given VAE across our three evaluation pillars, as illustrated in Figure 2. An input medical image $\mathbf{x}$ (either 2D or 3D) is passed through the VAE's encoder to produce a latent representation $\mathbf{z}$ and a reconstructed image $\hat{\mathbf{x}}$. These outputs are then systematically evaluated according to our three pillars: **Pillar 1 (Reconstruction Fidelity)** quantifies pixel-level accuracy using PSNR and SSIM. **Pillar 2 (Clinical Structure Preservation)** applies only to 3D data, where we use segmentation-based quality and similarity measures to assess structural plausibility. **Pillar 3 (Latent Representation Power)** evaluates the utility of $\mathbf{z}$ for downstream classification tasks, with performance compared against image-based baselines. Representative qualitative examples of reconstruction and subtraction maps are provided in Figure 3, demonstrating how these pillars complement each other in practice. For all metrics, let $\mathbf{x}$ be the original image and $\hat{\mathbf{x}}$ be the reconstructed image.

**Pillar 1: Reconstruction Fidelity.** This pillar assesses the global quality of the reconstructed image. For both 2D and 3D data, we report two metrics. First, the Peak Signal-to-Noise Ratio (PSNR),

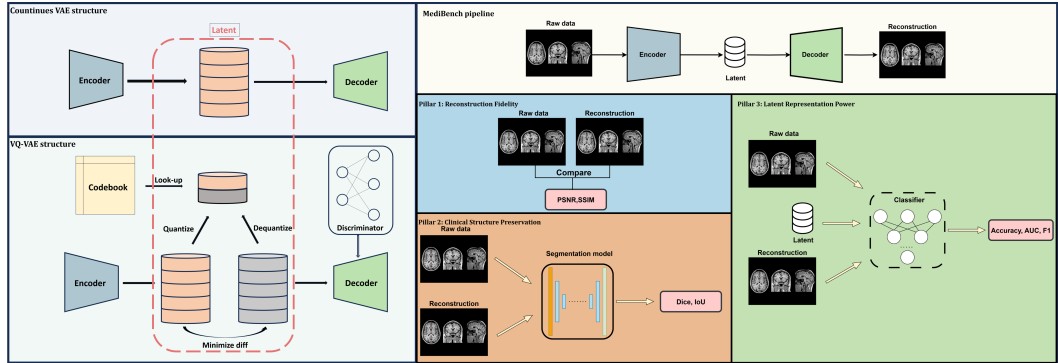

Figure 2: The MediBench evaluation pipeline. An input medical image **x** is processed to generate a latent representation **z** and a reconstructed image **x̂**. These outputs are then systematically assessed by our three evaluation pillars, corresponding to the dual mandate of understanding and generation. The pipeline is applied to multiple datasets across various modalities.

which measures the ratio between the maximum possible power of a signal and the power of corrupting noise that affects the fidelity of its representation:

$$\text{PSNR}(\mathbf{x}, \hat{\mathbf{x}}) = 10 \cdot \log_{10}\left(\frac{\text{MAX}_I^2}{\text{MSE}(\mathbf{x}, \hat{\mathbf{x}})}\right), \tag{3}$$

where $\text{MAX}_I$ is the maximum possible pixel value of the image and MSE is the Mean Squared Error. Second, the Structural Similarity Index (SSIM), which measures the similarity between two images based on luminance, contrast, and structure. Higher values indicate better similarity.

**Pillar 2: Clinical Structure Preservation.** This pillar evaluates whether diagnostically critical features are preserved, a key aspect of the generative mandate. It applies only to 3D data, since anatomical structures in volumetric scans (e.g., brain regions, organs) are meaningful to assess in this way. We leverage an automated segmentation tool as a proxy for expert anatomical assessment. First, Segmentation Quality Control (Seg QC) uses SynthSeg v2 Billot et al. (2023) to assign plausibility scores for reconstructed volumes. The final score is the average QC score across all anatomical regions of interest (ROIs). Second, Segmentation Similarity (SegSim) directly measures the voxel-wise label agreement between segmentation maps of the original and reconstructed images:

$$\text{SegSim}(\mathbf{x}, \hat{\mathbf{x}}) = \frac{1}{N}\sum_{i=1}^{N}\mathbf{1}(S(\mathbf{x})_i = S(\hat{\mathbf{x}})_i), \tag{4}$$

where $S(\cdot)$ denotes segmentation maps and $\mathbf{1}(\cdot)$ is the indicator function. Scores closer to 1 indicate better structural preservation.

**Pillar 3: Latent Representation Power.** This pillar quantifies the utility of the latent space for downstream understanding tasks by measuring the performance of classifiers trained directly on latent vectors. For 2D tasks, we report Accuracy, Macro F1-score, Precision, Recall, and Macro AUC. These metrics are compared against two image-based baselines: a ResNet-18 feature extractor with an MLP, and a simpler MLP trained directly on raw images. For 3D tasks, we report Image Accuracy (using a SEResNet152 from MonAI Cardoso et al. (2022) + MLP baseline trained on raw volumes) and Latent Accuracy (using an MLP classifier trained on the VAE's latent vectors).

## 4 EXPERIMENTS

### 4.1 DATASETS

We evaluate MediBench on a diverse suite of public datasets spanning histopathology, dermatoscopy, fundus photography, chest X-ray, and 3D magnetic resonance imaging(MRI)/computed tomography(CT). When official splits exist, we follow them and report on the official test set. Otherwise we construct patient/subject-wise splits to prevent leakage and report on the held-out validation set. Expanded dataset narratives and exact split/count details are provided in Appendix B.

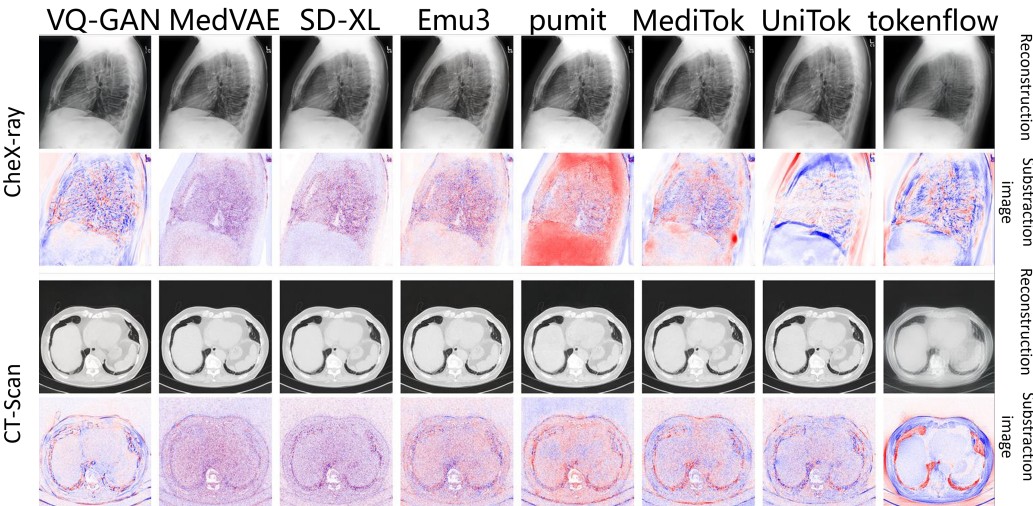

Figure 3: **Qualitative comparison of reconstructions and subtraction maps.** The main text shows two modalities (*CT scan*, *Chest X-ray*). Rows are modalities and columns are VAEs (MedVAE, MediTok, PUMIT, VQ-GAN, SDXL, TokenFlow, UniTok, Emu3). Top: reconstructions; Bottom: subtraction maps vs. ground truth. The complete panel appears in Appendix C Figure 6.

## 4.2 Experiment details and Results

We evaluate MediBench under a unified experimental skeleton and report consolidated findings. Implementation choices are standardized across models when possible and follow official releases. Full details, including the dataset overview tables formerly in the main text, now appear in Appendix.

**Qualitative reconstructions.** Figure 3 shows two representative 2D modalities. We include CT scan and chest X-ray to highlight challenging settings. (The full multi-modality panel is in Appendix.) Tokenized and VQ-style models (MediTok, PUMIT, TokenFlow, UniTok, Emu3, VQ-GAN) preserve sharper boundaries and leave fewer residuals in subtraction maps. Continuous VAEs (MedVAE, SDXL) produce smoother images but may attenuate fine details. These visual trends align with the quantitative results below.

**2D results.** Tokenization dominates latent classification. MediTok and PUMIT achieve very strong AUC on histopathology and dermatoscopy (e.g., 0.961 and 0.962 on histopathology; 0.961 and 0.923 on dermatoscopy in Table 2). TokenFlow is also competitive across modalities (column-average AUC 0.829). Continuous VAEs trail on AUC despite strong SSIM and PSNR. For example, SDXL has high SSIM in Table 1 but moderate AUC in Table 2. This shows that pixel fidelity and discriminative utility are weakly aligned in 2D.

**Resolution effects are modest.** Higher resolution improves SSIM and PSNR but often yields small AUC gains. Some datasets benefit slightly. Others do not. Ranking and thresholding can diverge on imbalanced tasks. Several models reach high AUC with low macro-F1. This indicates probability clustering near decision thresholds and weaker minority-class calibration.

**3D results.** Medically pre-trained tokenizers transfer well to volumes. On fMRI (Parkinson), MediTok reaches latent AUC 0.885 versus MedVAE 0.625 in Table 5. On CT (Coltea), MediTok attains latent AUC 0.983. Continuous decoders can deliver very high pixel fidelity. SDXL and MediTok both exceed 0.96 SSIM on CT (Table 3). Latent AUC does not always follow fidelity. Segmentation proxies are sensitive to structure. On T1 MRI (Parkinson), MediTok improves SegSim (99.2) over MedVAE (91.4) with similar SegQC (84.1 vs. 84.5) in Table 4. This suggests that large-scale pre-training can compensate for architectural constraints. It also explains why MediTok, although not natively 3D, is competitive on structure preservation.

Domain mismatch hurts continuous VAEs on specific cohorts. MedVAE underperforms on fMRI and multi-modal MRI in Table 3 (e.g., 0.359 SSIM on MRI Parkinson; 0.180 on BraTS). A likely

Table 1: **2D Panel A: Reconstruction fidelity**. psnr/ssim at $256\times256$. Columns are grouped by model family. The last column is the row average. The last row is the column average. Bold marks the maximum per column by SSIM.

| | Continuous VAEs | | VQ VAEs | | | | | | Avg |
|---|---|---|---|---|---|---|---|---|---|
| Modality | SDXL | MedVAE | VQ-GAN | MediTok | PUMIT | TokenFlow | UniTok | Emu3 | |
| Histopathology | 26.0/**0.740** | 20.6/0.840 | 19.5/0.394 | 25.6/0.796 | 20.9/0.773 | 21.1/0.461 | 24.5/0.718 | 24.3/0.678 | 22.8/0.675 |
| Dermatoscopy | 37.2/**0.905** | 22.5/0.905 | 29.4/0.769 | 33.6/0.899 | 29.0/0.881 | 31.1/0.789 | 33.5/0.869 | 34.9/0.865 | 31.4/0.860 |
| Fundus | 41.0/**0.964** | 20.5/0.900 | 30.1/0.821 | 36.6/0.958 | 33.9/0.937 | 33.0/0.905 | 36.4/0.941 | 37.8/0.944 | 33.7/0.921 |
| Chest X-ray | 30.5/0.881 | **34.2**/0.951 | 22.8/0.620 | 29.4/0.886 | 29.3/0.877 | 24.8/0.696 | 17.2/0.501 | 28.9/0.826 | 27.2/0.780 |
| **Average** | 33.7/0.872 | 24.5/**0.899** | 25.5/0.651 | 31.3/0.885 | 28.3/0.867 | 27.5/0.713 | 27.9/0.757 | 31.5/0.829 | N/A |

Table 2: **2D Panel B: Latent representation power**. AUC at $256\times256$. Columns are grouped by model family. Each row average is in the last column. The last row reports column-wise averages across modalities. Bold marks the maximum per column.

| | Continuous VAEs | | VQ VAEs | | | | | | Avg |
|---|---|---|---|---|---|---|---|---|---|
| Modality | SDXL | MedVAE | VQ-GAN | MediTok | PUMIT | TokenFlow | UniTok | Emu3 | |
| Histopathology | 0.842 | 0.500 | 0.856 | **0.961** | 0.962 | 0.952 | 0.889 | 0.769 | 0.841 |
| Dermatoscopy | 0.873 | 0.771 | 0.862 | **0.961** | 0.923 | 0.933 | 0.879 | 0.675 | **0.847** |
| Fundus | 0.681 | 0.611 | 0.636 | **0.883** | 0.817 | 0.753 | 0.701 | 0.702 | 0.723 |
| Chest X-ray | 0.565 | 0.578 | 0.638 | 0.659 | 0.650 | **0.677** | 0.655 | 0.631 | 0.631 |
| **Average** | 0.740 | 0.615 | 0.748 | **0.866** | 0.838 | 0.829 | 0.781 | 0.694 | N/A |

Table 3: **3D Panel A: Reconstruction fidelity**. psnr/ssim. Grouped by model family. Row and column averages included. Bold marks the maximum per column by SSIM.

| | Continuous VAEs | | | | VQ VAEs | | Avg |
|---|---|---|---|---|---|---|---|
| Modality | SDXL | MedVAE | VideoVAE | CVVAE | MediTok | CogVideoX | |
| fMRI (Parkinson) | 29.0/0.630 | 27.0/0.555 | 28.0/0.558 | 31.4/**0.776** | 30.1/0.784 | 27.5/0.712 | 28.8/0.669 |
| fMRI (NeuroEmo) | 43.1/**0.952** | 30.0/0.597 | 42.9/0.910 | 42.9/0.899 | 44.1/0.948 | 29.8/0.557 | 38.8/0.811 |
| MRI (Parkinson) | 45.6/**0.928** | 25.3/0.359 | 42.0/0.905 | 37.4/0.893 | 47.3/0.926 | 38.9/0.896 | 39.4/0.818 |
| MRI (BraTS 2023) | 26.0/0.204 | 23.5/0.180 | 22.3/0.158 | 17.8/0.150 | 40.7/**0.968** | 26.5/0.216 | 26.1/0.313 |
| CT (Coltea) | 41.1/**0.961** | 27.8/0.774 | 39.1/0.954 | 37.7/0.929 | 37.9/0.960 | 34.7/0.912 | 36.4/0.915 |
| **Average** | 37.0/0.735 | 26.7/0.493 | 34.9/0.697 | 33.4/0.729 | 39.4/**0.917** | 31.5/0.659 | N/A |

Table 4: **3D Panel B: Clinical structure preservation**. SegQC/SegSim. Grouped by model family. Row and column averages included. Bold marks the maximum per column.

| | Continuous VAEs | | | | VQ VAEs | | Avg |
|---|---|---|---|---|---|---|---|
| Modality | SDXL | MedVAE | VideoVAE | CVVAE | MediTok | CogVideoX | |
| MRI (Parkinson) | **99.1**/84.0 | 84.5/91.4 | 98.3/83.5 | 98.5/83.4 | 84.1/**99.2** | 84.2/**99.2** | **91.5**/90.1 |
| MRI (BraTS 2023) | 79.5/98.3 | 79.4/98.1 | 70.8/95.7 | 78.6/97.6 | **79.6**/98.3 | 79.2/**98.5** | 77.9/97.8 |
| **Average** | **89.3**/91.2 | 82.0/94.8 | 84.6/89.6 | 88.6/90.5 | 81.9/**98.8** | 81.7/**98.9** | N/A |

cause is limited exposure to those contrasts during pretraining. Pretraining scale matters. MediTok benefits from broader pretraining data and thus outperforms MedVAE on several volumetric tasks, including improved SegSim (98.8 vs. 94.8) and similar SegQC (81.9 vs. 82.0). Tokenized and VQ-style models (MediTok, CogVideoX) also preserve more clinical structure detail.

Latent size and compression. Table 6 reports latent shapes and compression ratios for $256^2$ and $512^2$ inputs and a representative 3D setting. Combining these with publicly available parameter counts where checkpoints are disclosed, we observe that raw parameter count is a weak predictor of downstream performance. Tokenizer type and latent compactness matter more. Tokenized and VQ models (MediTok, PUMIT, TokenFlow) reach the highest latent AUCs in 2D with compact latents (e.g., MediTok uses 768 tokens, ratio 0.004 at $256^2$ and 0.001 at $512^2$). Continuous VAEs such as SDXL-VAE have ratios around 0.083. High SSIM and PSNR do not guarantee strong latent utility. SDXL shows high reconstruction fidelity and moderate AUC on several datasets. This matches

Table 5: **3D Panel C: Latent representation power**. AUC for latent (top) and reconstruction-embedding (bottom). Grouped by model family. Row and column averages included. Bold marks the maximum per column.

| | Continuous VAEs | | | | VQ VAEs | | Avg |
|---|---|---|---|---|---|---|---|
| **Latent Embedding** | SDXL | MedVAE | VideoVAE | CVVAE | MediTok | CogVideoX | |
| fMRI (Parkinson) | **0.985** | 0.625 | 0.926 | 0.910 | 0.885 | 0.465 | **0.799** |
| fMRI (NeuroEmo) | 0.506 | 0.519 | 0.531 | **0.553** | 0.493 | 0.501 | 0.517 |
| MRI (Parkinson) | **0.560** | 0.415 | 0.535 | 0.526 | 0.485 | 0.551 | 0.512 |
| CT (Coltea) | 0.801 | 0.592 | 0.776 | 0.714 | **0.983** | 0.814 | 0.780 |
| **Average** | **0.713** | 0.538 | 0.692 | 0.676 | 0.709 | 0.583 | N/A |
| **Reconstruction Image Embedding** | SDXL | MedVAE | VideoVAE | CVVAE | MediTok | CogVideoX | |
| fMRI (Parkinson) | **0.580** | 0.482 | 0.508 | 0.509 | 0.575 | 0.547 | 0.533 |
| fMRI (NeuroEmo) | **0.512** | 0.496 | 0.507 | 0.510 | 0.510 | 0.493 | 0.505 |
| CT (Coltea) | 0.625 | 0.667 | 0.620 | **0.767** | 0.683 | 0.687 | **0.675** |
| **Average** | 0.572 | 0.548 | 0.545 | **0.595** | 0.589 | 0.576 | N/A |

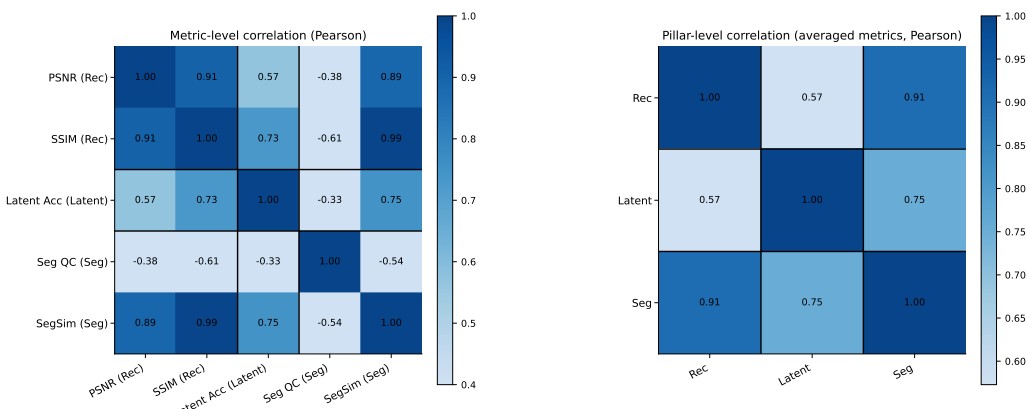

Figure 4: **Correlation heatmaps on the 3D MRI T1 dataset.** Left: metric-level correlation among PSNR, SSIM, latent accuracy, and segmentation metrics. Right: pillar-level correlation. Both heatmaps are block-partitioned by downstream tasks. Black gridlines delineate task boundaries.

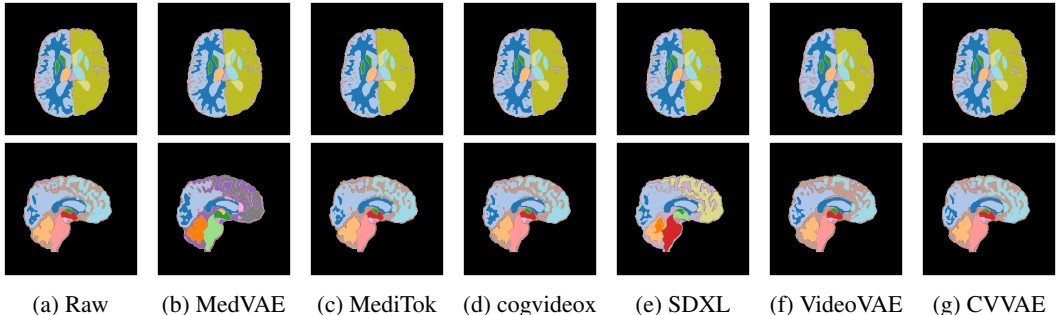

| (a) Raw | (b) MedVAE | (c) MediTok | (d) cogvideox | (e) SDXL | (f) VideoVAE | (g) CVVAE |
|---|---|---|---|---|---|---|

Figure 5: Segmentation overlays for reconstructed 3D scans (axial and sagittal views).

the finding that pixel fidelity and discriminative power are only partially aligned. In 3D, medical pretraining and tokenizer design dominate. MediTok improves structure-preservation metrics over MedVAE with similar quality-control scores. It also achieves strong latent AUC on CT although it is not a native 3D encoder.

Table 6: **Latent shapes and compression ratios**. Shapes from representative runs. Compression ratio is #latent elements divided by #input elements. 2D inputs use 3 channels. Lower is more compact.

| Method | Latent Shape | #Elements | Ratio @$256^2$ (@$512^2$) |
|---|---|---|---|
| VQ-GAN | (256, 32, 32) | 262,144 | 1.333 (0.333) |
| MedVAE | (3, 128, 128) | 49,152 | 0.250 (0.063) |
| SDXL-VAE | (4, 64, 64) | 16,384 | 0.083 (0.021) |
| Emu3-VQ | (4, 1, 64, 64) | 16,384 | 0.083 (0.021) |
| PUMIT | (512, 32, 32) | 524,288 | 2.667 (0.667) |
| MediTok | (768) | 768 | 0.004 (0.001) |
| UniTok | (1024, 1024) | 1,048,576 | 5.333 (1.333) |
| TokenFlow | (40, 32, 32) | 40,960 | 0.208 (0.052) |
| *3D example* (input $128{\times}128{\times}36$; total elements 589,824) | | | |
| MedVAE | (32, 32, 9) | 9,216 | 0.0156 |
| MediTok | (36, 768) | 27,648 | 0.0469 |
| CogVideoX | (16, 32, 16, 4) | 32,768 | 0.0556 |
| SDXL | (36, 4, 16, 16) | 36,864 | 0.0625 |
| VideoVAE | (4, 9, 16, 16) | 9,216 | 0.0156 |
| CVVAE | (4, 10, 16, 16) | 10,240 | 0.0174 |

## 5 CONCLUSION AND DISCUSSION

This work introduces MediBench, a benchmark for medical VAEs that measures reconstruction fidelity, clinical structure preservation, and latent utility in a single framework. We evaluate continuous and tokenized architectures across diverse modalities and tasks. The analysis yields three central findings. First, tokenized and vector quantized encoders provide stronger latent representations than continuous VAEs. They preserve sharp boundaries and fine textures in reconstructions. They also produce latents that support classification across multiple datasets. MediTok, PUMIT, and Token-Flow achieve the highest AUC in most 2D settings. They remain competitive in 3D tasks. Second, medical pretraining matters. Models with broader exposure to clinical data preserve anatomy more faithfully and transfer better. Continuous VAEs trained on natural images can attain high SSIM and PSNR. They do not always yield high downstream performance. The gap is largest under domain shift, such as fMRI and multi–sequence MRI. MediTok benefits from larger and more diverse pretraining and performs well even without a native 3D encoder. Third, pixel fidelity and discriminative utility are only partially aligned. Raising resolution from 256 to 512 increases SSIM and PSNR. AUC often changes little and sometimes drops. High AUC with low macro–F1 is common on imbalanced tasks. Scores rank classes well yet cluster near default thresholds. Calibration and class–aware decision rules are therefore important.

Our study also highlights the value of structure–aware evaluation. Segmentation–derived metrics are more sensitive to small boundary shifts than pixel metrics. They better reflect clinical plausibility. They capture failure modes that SSIM and PSNR may miss.

**Limitations.** This benchmark has limitations. The datasets are diverse yet finite. The CheXpert split is a curated subset. SLAKE relies on VQA–style labels that are not dense segmentations. Some 3D evaluations adapt models trained primarily on 2D data. Segmentation proxies do not replace clinical endpoints. We report consolidated results in the main paper and provide extensive tables in the appendix.

**Future work** will expand both scope and depth. We will add ultrasound and PET and include multi–institution cohorts. We will study calibration, robustness, and fairness under shift. We will incorporate native 3D tokenizers and cross–resolution consistency tests. We will evaluate clinical endpoints and decision support scenarios. We will release task–specific adapters and stronger segmentation oracles. These steps will advance the selection and design of medical VAEs and will improve the reliability of latent spaces used in clinical pipelines.

ETHICS STATEMENT

This work uses only publicly available medical-imaging datasets under their respective licenses. All data were de-identified by the providers prior to our access. No additional data collection or interaction with human subjects was conducted; therefore, no new IRB approval was required. We report results at the subject level where applicable and avoid leakage via patient-wise splits.

We consider potential risks, including privacy leakage and clinically misleading reconstructions. We mitigate these risks by (i) using patient-wise splits and standardized preprocessing, (ii) auditing reconstructions with structure-preservation and quality-control metrics, and (iii) releasing only the artifacts needed to reproduce the reported tables and figures under a research license. We discuss dual-use concerns (e.g., synthetic images) and restrict releases to research purposes.

We estimate the computational footprint of training and evaluation in the Appendix and encourage energy-efficient replication and reporting of compute/energy usage in follow-up work.

USE OF GENERATIVE AI

We used a large language model *only* to aid writing (grammar/style polishing) after the technical content was drafted by the authors. The model was *not* used to generate datasets, code, results, analyses, or claims. All scientific content, tables, and numbers originate from our pipeline and were manually verified by the authors. Texts revised by generative AI are manually reviewed to ensure factual correctness and to avoid biased or fabricated statements.

REPRODUCIBILITY STATEMENT

We release code, configs, and scripts to reproduce every table and figure in the paper.[1] Our release includes:

- **Datasets and splits.** Exact dataset versions, licenses, and subject-wise splits; preprocessing steps with parameters (see Section 4 and Appendix B).
- **Environments.** Hardware/software details (GPU type, CUDA/cuDNN, Python/PyTorch), dependency lockfiles, and Docker files; fixed random seeds (see Appendix A).
- **Training and inference.** Commands and configuration files for each model; hyperparameters (optimizer, schedule, batch size, epochs), and expected wall-clock time (see Section 4 and Appendix A).
- **Evaluation.** Scripts for reconstruction metrics (e.g., SSIM/PSNR), structure-preservation and quality-control metrics, and latent-utility tasks with identical protocols across models (see Section 3).
- **Models.** Pointers to official checkpoints where licenses permit; for each VAE/tokenizer, we document any deviations from the authors' official recipes.
- **Determinism.** We set seeds and log non-deterministic CuDNN flags; we report mean $\pm$ std over multiple runs where appropriate (see Appendix A).

These materials allow independent researchers to reproduce the reported numbers and regenerate all plots and tables with a single command per experiment.

---

[1]Repository and checkpoints: `<anonymized-link>`.

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

## A EXPERIMENTAL DETAILS

**Environments.** All 2D experiments use a single NVIDIA RTX A5500 (24GB). All 3D experiments use a single NVIDIA RTX 4090. For each VAE (2D/3D) we adhere to its *official* environment (library versions, CUDA/cuDNN, dependencies) as specified by the authors to ensure out-of-the-box behavior; custom code is limited to data I/O, metric computation, and training wrappers.

**Pre-processing.** For 2D data, images are resized to 256×256 and 512×512, converted to 3-channel RGB, then normalized following each VAE's official recipe. For 3D volumes, intensities are normalized to $[0, 1]$ for I/O and to $[-1, 1]$ at the VAE encoder input; reconstructions are denormalized to $[0, 1]$ for metrics.

**Downstream classifiers.** For 2D, we train an MLP on latents and compare against two image baselines (ResNet-18+MLP; MLP on raw images). For 3D, the image baseline uses SEResNet152 features (2048-d) with a linear head; the latent branch flattens the latent and applies a linear classifier. Optimizers, LR schedules, batch sizes and early stopping follow Section 4; any remaining dataset-specific settings are provided alongside the full tables below.

**Segmentation-driven evaluation (3D).** We compute segmentation maps on originals and reconstructions and report quality/similarity measures (SegQC/SegSim). These metrics reflect preservation of semantically meaningful anatomy beyond pixel similarity.

## B DATASET DETAILS

**CheXpert-Plus (subset).** A curated subset of 5,000 frontal CXR with 14 thoracic findings; strict patient-wise 80/20 split (4,000/1,000), reporting on the held-out test.

**SLAKE (VQA-derived classification).** This is a VQA corpus where each image is paired with Q/A tuples, e.g., {"question": "What modality is used?", "answer": "MRI", "content_type": "Modality"} or {"question": "Which part of the body?", "answer": "Abdomen", "content_type": "Position"}. We derive a 3-class *content-type* label ({Modality, Organ, Position}) from these annotations, enabling a clinically meaningful metadata classification without pixel-level labels.

**Other datasets.** MedMNIST v3 subsets (Path/Derma/Retina) use official splits; PCam uses official splits; DERM12345 uses patient-wise 80/20; DDR follows the official split (13,673 train / 6,667 test). 3D cohorts (ds005700, ds005892, Coltea, BraTS 2023) follow the subject-wise protocols summarized in Tables 11–12 (main text).

## C   FULL QUALITATIVE PANELS

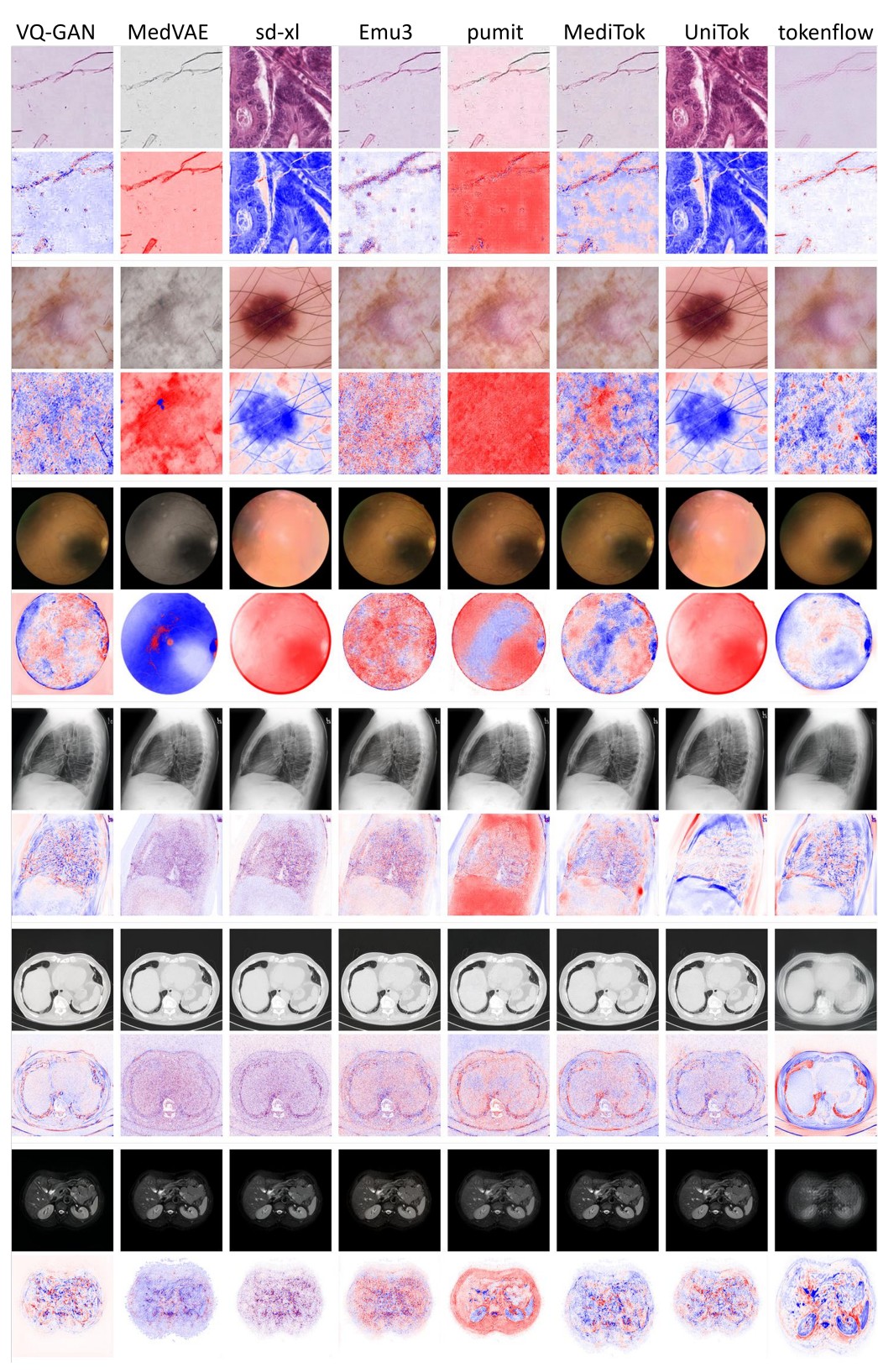

Figure 6: **Complete modality-organized panel of reconstructions and subtraction maps across all VAEs.** This expands Figure 3 in the main text to include all modalities and all methods.

Table 7: **2D Reconstruction (256×256)**. SSIM and PSNR dB per dataset and method.

| Dataset | MedVAE | MediTok | PUMIT | VQ-GAN | SDXL | TokenFlow | UniTok | Emu3 |
|---|---|---|---|---|---|---|---|---|
| dermamnist | 0.905 (22.5) | 0.899 (33.6) | 0.881 (29.0) | 0.769 (29.4) | 0.905 (37.2) | 0.789 (31.1) | 0.869 (33.5) | 0.865 (34.9) |
| pathmnist | 0.840 (20.6) | 0.796 (25.6) | 0.773 (20.9) | 0.394 (19.5) | 0.740 (26.0) | 0.461 (21.1) | 0.718 (24.5) | 0.678 (24.3) |
| retinamnist | 0.900 (20.5) | 0.958 (36.6) | 0.937 (33.9) | 0.821 (30.1) | 0.964 (41.0) | 0.905 (33.0) | 0.941 (36.4) | 0.944 (37.8) |
| pcam | 0.898 (21.6) | 0.858 (25.2) | 0.848 (29.8) | 0.379 (17.6) | 0.796 (25.0) | 0.464 (19.1) | 0.772 (23.8) | 0.718 (23.0) |
| ddr | 0.818 (19.7) | 0.903 (35.1) | 0.880 (32.6) | 0.762 (29.1) | 0.916 (38.2) | 0.805 (29.6) | 0.850 (32.3) | 0.887 (35.6) |
| chexpert_plus | 0.951 (34.2) | 0.886 (29.4) | 0.877 (29.3) | 0.620 (22.8) | 0.881 (30.5) | 0.696 (24.8) | 0.501 (17.2) | 0.826 (28.9) |
| slake | 0.964 (36.1) | 0.926 (30.8) | 0.906 (29.8) | 0.731 (23.2) | 0.926 (32.2) | 0.817 (25.6) | 0.905 (29.5) | 0.901 (30.6) |

Table 8: **2D Reconstruction (512×512)**. SSIM and PSNR dB per dataset and method.

| Dataset | MedVAE | MediTok | PUMIT | VQ-GAN | SDXL | TokenFlow | UniTok | Emu3 |
|---|---|---|---|---|---|---|---|---|
| dermamnist | 0.947 (22.8) | 0.956 (35.1) | 0.910 (26.6) | 0.879 (31.1) | 0.962 (40.3) | 0.867 (30.5) | 0.948 (36.9) | 0.940 (37.3) |
| pathmnist | 0.919 (21.9) | 0.917 (29.8) | 0.877 (22.2) | 0.596 (22.8) | 0.911 (32.2) | 0.524 (21.0) | 0.889 (29.6) | 0.849 (29.6) |
| retinamnist | 0.922 (20.7) | 0.978 (39.0) | 0.950 (33.5) | 0.826 (33.5) | 0.983 (44.3) | 0.883 (28.4) | 0.971 (39.0) | 0.971 (40.2) |
| pcam | 0.939 (21.4) | 0.958 (31.2) | 0.897 (30.2) | 0.621 (21.8) | 0.968 (34.3) | 0.530 (19.2) | 0.941 (30.4) | 0.915 (30.1) |
| ddr | 0.761 (16.5) | 0.878 (35.6) | 0.858 (31.9) | 0.748 (31.5) | 0.895 (38.2) | 0.741 (26.0) | 0.831 (32.8) | 0.866 (36.0) |
| chexpert_plus | 0.943 (35.7) | 0.888 (31.0) | 0.856 (26.9) | 0.688 (25.2) | 0.884 (32.7) | 0.664 (23.7) | 0.531 (17.2) | 0.841 (31.1) |
| slake | 0.965 (39.7) | 0.932 (33.7) | 0.908 (29.2) | 0.755 (26.4) | 0.928 (35.8) | 0.716 (23.6) | 0.913 (32.5) | 0.910 (33.3) |

# D  FULL PER-DATASET TABLES (256 AND 512)

## D.1  2D RECONSTRUCTION FIDELITY (PER DATASET, ALL METHODS)

Each cell reports SSIM (PSNR in dB). Results are shown separately for 256×256 and 512×512.

## D.2  CHEXPERT-PLUS LATENT CLASSIFICATION (256 AND 512)

Metrics include exact-match Accuracy, Macro AUC, Macro F1, Macro Precision, and Macro Recall. We report latent-based classifiers trained on the VAE representations for both 256×256 and 512×512.

# E  OVERVIEW OF DATASETS

Table 9: **CheXpert-Plus latent classifiers** at 256×256. Metrics are Accuracy, Macro AUC, Macro F1, Precision, Recall (per method).

| Method | Acc. | AUC | F1 | Prec. | Rec. |
|---|---|---|---|---|---|
| MedVAE | 0.738 | 0.578 | 0.061 | 0.053 | 0.071 |
| MediTok | 0.738 | 0.659 | 0.061 | 0.056 | 0.068 |
| PUMIT | 0.739 | 0.650 | 0.056 | 0.051 | 0.064 |
| VQ-GAN | 0.739 | 0.638 | 0.056 | 0.051 | 0.064 |
| SDXL | 0.740 | 0.565 | 0.056 | 0.049 | 0.065 |
| TokenFlow | 0.740 | 0.677 | 0.063 | 0.056 | 0.073 |
| UniTok | 0.740 | 0.655 | 0.058 | 0.052 | 0.068 |
| Emu3 | 0.740 | 0.631 | 0.061 | 0.053 | 0.071 |

Table 10: **CheXpert-Plus latent classifiers** at 512×512. Metrics are Accuracy, Macro AUC, Macro F1, Precision, Recall (per method).

| Method | Acc. | AUC | F1 | Prec. | Rec. |
|---|---|---|---|---|---|
| MedVAE | 0.741 | 0.582 | 0.061 | 0.053 | 0.071 |
| MediTok | 0.741 | 0.661 | 0.061 | 0.056 | 0.068 |
| PUMIT | 0.742 | 0.653 | 0.056 | 0.051 | 0.064 |
| VQ-GAN | 0.742 | 0.641 | 0.056 | 0.051 | 0.064 |
| SDXL | 0.742 | 0.567 | 0.056 | 0.049 | 0.065 |
| TokenFlow | 0.743 | 0.680 | 0.063 | 0.056 | 0.073 |
| UniTok | 0.742 | 0.657 | 0.058 | 0.052 | 0.068 |
| Emu3 | 0.742 | 0.634 | 0.061 | 0.053 | 0.071 |

Table 11: Overview of **2D** datasets in MediBench. We follow official splits when available; otherwise we use patient-wise splits. Tasks are expressed in clinically meaningful terms (disease/severity/lesion) rather than generic "cls.".

| Dataset | Modality (Input) | Task | Train/Total |
|---|---|---|---|
| PathMNIST (MedMNIST v3) | Histopathology (224×224) | Colon tissue-type classification (9 classes) | 89,996 / 107,180 |
| DermaMNIST (MedMNIST v3) | Dermatoscopy (224×224) | Skin lesion classification (7 classes) | 7,007 / 10,015 |
| RetinaMNIST (MedMNIST v3) | Fundus (224×224) | Diabetic retinopathy severity grading (5 ordinal classes) | 1,080 / 1,600 |
| PatchCamelyon (PCam) | Histopathology (96×96) | Metastatic tumor detection (binary) | 262,144 / 327,680 |
| DERM12345 | Dermatoscopy ($\rightarrow$ 224×224) | Fine-grained skin lesion classification (40 classes) | 9,876 / 12,345 |
| DDR | Fundus ($\rightarrow$ 224×224) | Diabetic retinopathy grading (5 classes) | 13,673 / 20,340 |
| CheXpert-Plus | Chest X-ray ($\rightarrow$ 224×224) | Thoracic disease multi-label classification (14 labels) | 4,000 / 5,000 |
| SLAKE | Mixed CT/MRI/X-ray ($\rightarrow$ 224×224) | Content-type classification (3 classes: Modality / Organ / Position) from VQA annotations | 513 / 642 |

Table 12: Overview of **3D** datasets in MediBench. Custom splits are subject-wise unless specified. Tasks are phrased as disease/structure–centric semantics.

| Dataset | Modality (Input) | Task | Split / Size |
|---|---|---|---|
| ds005892 (functional) | fMRI | Parkinson's disease status classification (Healthy vs. PD vs. PD-MCI) | 6 subjects held out; 49 / 55 subjects |
| ds005700 | fMRI $\rightarrow$ 3D (128×128×36) | Emotion recognition classification (6 categories) | 4 subjects held out; 36 / 40 subjects |
| Coltea-/Lung-/CT-/100W | Triphasic CT ($\sim$ 512×512×350) | Contrast phase classification (Non-contrast, Venous, Arterial) | 10 subjects held out; 90 / 100 subjects |
| ds005892 (structural) | T1w MRI | Parkinson's disease status classification (Healthy vs. PD vs. PD-MCI) | 10-fold CV; 49 / 55 subjects |
| ds005892 (structural) | T1w MRI | Brain tissue segmentation (grey matter, cerebellum, etc.) | |
| BraTS 2023 | Multi-modal MRI (240×240×155) | Brain tumor segmentation (enhancing, core, whole tumor) | |

