# OpenReview forum: "MediBench: A Benchmark for VAEs in Medical Imaging Across Fidelity, Structure, and Latent Utility"
_ICLR.cc/2026/Conference — Submitted to ICLR 2026_

### Official Review · Reviewer_yDvX · 2025-10-16

**Soundness:** 3
**Presentation:** 2
**Contribution:** 2
**Rating:** 4
**Confidence:** 3

**Summary:**

The paper introduces MediBench, a benchmark for evaluating VAEs and VQ-VAEs in medical imaging across three aspects: reconstruction fidelity, clinical structure preservation, and latent space utility. Experiments on diverse 2D and 3D datasets show that tokenized and medically pre-trained VAEs achieve stronger latent representations, while high pixel fidelity does not necessarily improve downstream performance. MediBench offers a standardized framework for assessing and comparing medical generative models.

**Strengths:**

**Clear three-phase evaluation design:** Proposes a three-phase strategy that separately evaluates reconstruction fidelity, structural preservation, and latent utility within a unified 2D/3D multimodal framework.

**Comprehensive experimental evaluation:** The benchmark is tested across multiple datasets and imaging modalities (2D and 3D), providing a broad and systematic comparison of VAE and VQ-VAE models.

**Reproducible and easy to apply:** Builds on established metrics within a standardized evaluation strategy, making the framework practical and convenient for consistent model comparison.

**Weaknesses:**

**Lack of dataset usage transparency:**
While the benchmark covers multiple 2D and 3D medical datasets, the paper does not clearly specify how each dataset is used — including subset selection, sample size, and preprocessing protocols. Providing these details would improve reproducibility and clarify the benchmark’s representativeness.

**Limited model diversity within the VAE family:**
Although the benchmark focuses on the VAE family, the experimental coverage remains narrow, mainly comparing standard VAE and VQ-VAE variants. A broader inclusion of diverse VAE formulations would provide a more comprehensive evaluation of the proposed framework.

**Limited metric novelty:**
While the three-phase evaluation framework is well-structured, its individual components (SSIM/PSNR for fidelity, segmentation-based metrics for structure, and AUC for latent utility) largely rely on existing measures. The paper argues that combining them reveals clinically meaningful trade-offs and failure modes overlooked by traditional metrics, but the framework’s methodological advantage remains at the organizational level rather than introducing fundamentally new evaluation metrics.

**Questions:**

1. Can the authors further clarify how the proposed evaluation metrics generalize beyond the specific VAE and VQ-VAE variants, to demonstrate the framework’s broader applicability across diverse VAE-based generative models?

2. Can the authors provide more detailed information about each benchmark dataset, including subset selection, sample size, and preprocessing or split protocols?

3. Since each evaluation phase reuses existing metrics (SSIM/PSNR, segmentation-based Dice, and AUC), could the authors clarify what unique advantage the proposed three-phase framework provides beyond combining these established measures?

---

### Official Review · Reviewer_scbW · 2025-10-21

**Soundness:** 2
**Presentation:** 2
**Contribution:** 1
**Rating:** 2
**Confidence:** 3

**Summary:**

In this work, authors introduce MediBench, a benchmark that is designed to evaluate the performance of existing VAE-based methods in the medical domain. In particular, authors evaluate such methods across three categories: Reconstruction Fidelity, Clinical Structure Preservation, and Latent Space Utility. The first two categories focus on assessing whether the clinically relevant structures are maintained in the reconstructions whereas the last category focuses on measuring the effectiveness of the latent space in terms of relevant downstream tasks. The authors evaluate general purpose and medical-specific VAE architectures across 2D and 3D modalities, on diverse medical modalities/datasets (including WSI, Dermatology, Ophthalmology, 3D Brain MRI, Chest X-ray and CT Scan). Authors observe a trade-off across the 3 “pillars” that they have defined. Specifically, tokenized and vector quantized VAEs appear to learn more useful latent features that are relevant for downstream tasks compared to their continuous counterparts. On a related note, authors note that high pixel fidelity does not necessarily translate into gains in downstream tasks.

**Strengths:**

* I appreciate the diverse list of modalities (WSI, Dermatology, Ophthalmology, 3D Brain MRI, CheX-ray, CT Scan) considered in this benchmark.
* The benchmark is evaluated on a comprehensive set of baselines including recent works such as MedVAE and MedITok.

**Weaknesses:**

* A major weakness of the works is that the "Related Work" section is lacking in terms of VAEs in the context of medical imaging. Particularly, the methods authors evaluate in this work such as MediTok, MedVAE, etc. are not cited nor discussed in this section. This makes it difficult to contextual this work and understand what exactly authors are evaluating in this work. MedITok and MedVAE has their own respective evaluation pipelines reporting reconstruction quality and performance of latents on downstream classification tasks. Authors should discuss why this work goes beyond the scope of what is already in these (and other) works in the literature. Also related to this, authors list SDXL as a continuous VAE (line 302) but SDXL is widely known as one of the popular general purpose latent diffusion models. It is not clear to me what is the connection of this model and medical image reconstruction.
* While the list of modalities is comprehensive, the type of datasets covered (especially for MRI for which I have more expertise) is not ideal. A big chunk of the MRI data belong to non-healthy subjects. Therefore, the coverage of the datasets considered in the benchmark is not sufficient in my opinion. This can be potentially alleviated by including some of the popular datasets such as fastMRI/Stanford 3D.
* Please also see the questions below.

**Questions:**

***Questions and Suggestions:***
* In Figure $1$, I would suggest flipping the text in the bottom half of the image to ease readability.
* Following up with Weakness 1: what does SDXL and SDXL-VAE refer to specifically in Table $6$?
* Following up with Weakness 2: Did authors consider adding fastMRI [1] dataset as part of the benchmark? It is one of the most popular and adopted datasets for benchmarking MRI reconstruction performance of models (see [2,3,4] as examples).
* Did the authors consider extending Pillar 2 to modalities outside of MRI? This is potentially do-able with an off-the-shelf segmentation model such as MedSAM [5]. I think such an extension would greatly increase the impact of this work.
* What is the "downstream understanding task" concretely? Specifically AUC numbers reported under Section 4.2 are based on which classification task?
* The MedVAE performance in Table $1$ is rather strange to me. The SSIM numbers are very strong but corresponding PSNR numbers are very low. Could the authors comment on why that might be the case (particularly because these two metrics are expected to be more or less correlated)? For instance comparing MedVAE  and TokenFlow for Fundus images, they both have SSIM of $\approx0.9$ but PSNR values are significantly different ($\approx20$ vs $\approx33$).

***
***Typos:***
* line 324 and 3: psnr/ssim -> PSNR/SSIM
* line 333: PSNR number is bolded although caption states SSIM are bold.

***

***References:***

[1] https://fastmri.med.nyu.edu/

[2] Sriram, Anuroop, et al. "End-to-end variational networks for accelerated MRI reconstruction." International conference on medical image computing and computer-assisted intervention. Cham: Springer International Publishing, 2020.

[3] Fabian, Zalan, Berk Tinaz, and Mahdi Soltanolkotabi. "Humus-net: Hybrid unrolled multi-scale network architecture for accelerated mri reconstruction." Advances in Neural Information Processing Systems 35 (2022): 25306-25319.

[4] Lin, Kang, and Reinhard Heckel. "Robustness of deep learning for accelerated MRI: benefits of diverse training data." arXiv preprint arXiv:2312.10271 (2023).

[5] https://github.com/bowang-lab/MedSAM

---

### Official Review · Reviewer_sUXW · 2025-11-01

**Soundness:** 2
**Presentation:** 3
**Contribution:** 2
**Rating:** 4
**Confidence:** 4

**Summary:**

This work presents a benchmark study -- Medibench -- focused on validating different variational autoencoders (VAEs) for their usefulness in medical image analysis (MIA). Authors use different VAEs and assess their performance using three classes of metrics focusing on: reconstruction fidelity, clinical structure preservation, and latent representation power. Experiments are reported using several VAEs on multiple 2d/3d datasets. They conclude that tokenized VAEs yield stronger latent representations, using pretraining aligned with task is useful,  and reconstruction quality is not aligned with clinical utility.

**Strengths:**

* Studying the limitations of VAEs for MIA in a large-scale setting, using broad array of methods, datasets, and metrics is an interesting contribution.
* Using three classes of metrics focusing on different aspects of VAE performance (reconstruction, clinical structure preservation, and latent representation power) is a useful way to focus on isolating the capabilities of VAEs.
* Paper is generally easy to follow, and nicely written.

**Weaknesses:**

* **Over-emphasis on VAEs:**

    > VAEs are now central to medical AI... (L74-80)

    This is disputable. What is this claim based on? Are there deployed medical systems that are reliant on VAEs? No references are provided to back these claims that VAEs are central to medical AI from a practical standpoint.

* **Missing Framework:** The authors state in several places that MediBench is a framework. This is misleading. I think the paper presents a series of benchmarking results on public datasets. The contribution primarily is the benchmarking of different VAEs, comparing them across three metrics. This is not a framework in the common understanding of the term.

* **Narrow focus:** Using VAEs for medical image analysis, and benchmarking them without being grounded in real-world outcomes is a weakness of the work. As pointed out already above, VAEs or latent generative models are not used in a widespread manner in clinical settings.

* **No clinically relevant measures:** The three measures focus on reconstruction and latent space usefulness. These are not directly tied to clinical outcomes. Latent representation utility is the metric closest to assessing downstream performance; this however, is seldom the application of VAEs that clinics embed high-resolution images and then use prediction based on the latent vectors. Furthermore, the class of models used to compare this performance (ResNet-18 feature extractor + MLP, and simple MLP) are not how medical image analysis models are used in clinical settings. Task-specific models are widely used, for example, workflows with nnUnet to perform segmentation of ROIs and then classification heads on top of these predictions are more common [1].

* **Cohort-level studies are missing:** The most useful scenario where VAEs could be used is in performing population/cohort level studies. These could be for unsupervised clustering, or for anomaly detection [2]. This work, however, focuses on image-level markers.

* **Methods not described:** In Sec 3.1, VAEs and VQ-VAEs are defined. However, the specific type of these models reported in the benchmark are not described anywhere (including in the Appendix). These are not common VAEs: MedVAE,MediTok, PUMIT, VQ-GAN, SDXL, TokenFlow, UniTok, Emu3, or at least not that I know very well.

* **Compression ratios:** For several methods reported in Table 6 (VQ-GAN,PUMIT, UniTok) the compression ratio is > 1, some even 5x more expensive. And these are some of the methods that do well in some of the metrics reported. So, how are these being efficient, which is the original motivation of arguing for VAEs in clinical settings?

### References

[1] Isensee, Fabian, et al. "nnU-Net: a self-configuring method for deep learning-based biomedical image segmentation." Nature methods 18.2 (2021): 203-211.

[2] Grønbech, Christopher Heje, et al. "scVAE: variational auto-encoders for single-cell gene expression data." Bioinformatics 36.16 (2020): 4415-4422.

**Questions:**

See weaknesses above.

---

### Official Review · Reviewer_HX5J · 2025-11-03

**Soundness:** 2
**Presentation:** 2
**Contribution:** 2
**Rating:** 2
**Confidence:** 3

**Summary:**

The paper introduces MediBench, a three-pillar benchmark for medical VAEs. It evaluates continuous, tokenized/VQ, and medically pretrained encoders across 2D/3D datasets. Findings: tokenized/VQ latents often outperform continuous VAEs for downstream tasks; medical pretraining improves transfer and structure retention; pixel fidelity correlates weakly with downstream performance.

**Strengths:**

1. Addresses a real gap: medical images are high-res and compute-hungry; a standardized way to assess compressive latent models is useful.
2. Three-pillar design tries to cover both generation and understanding, and it explicitly surfaces the fidelity–utility trade-off.
3. Breadth across modalities (2D/3D) and the finding that SSIM/PSNR don’t predict clinical utility is valuable.

**Weaknesses:**

Main Concerns:

1. The paper assumes VAEs are the right foundation for medical imaging without proving this against the strongest non-generative baselines (self-supervised discriminative encoders, masked autoencoders, supervised/ImageNet or medical-pretrained CNN/ViT backbones). If the community’s current practice skews discriminative, a benchmark restricted to VAEs risks being narrow and of limited external validity.

2. Many medical workflows only need compact features, not reconstructions. The paper should justify why a probabilistic generative criterion (ELBO/KL) is necessary vs. feature learners that optimize contrastive/masked-reconstruction objectives, or deterministic AEs. Otherwise, the benchmark may be optimizing the wrong abstraction for clinical tasks.

3. It’s unclear whether downstream results isolate representation quality or reflect extra modeling capacity/training on top of the latents.

- *Are downstream classifiers linear probes (frozen encoder) or fine-tuned? If fine-tuned, the comparison becomes muddier and favors larger heads.*

- *Are all methods compared at matched compute/params/wall-clock? Without cost–quality tradeoffs, it’s hard to argue for VAEs in practice.*

4. Protocol details are unclear in Pillar 2: was the segmenter trained only on raw images and evaluated on raw vs. recon to avoid leakage? As using a segmentation model on reconstructed images can entangle VAE artifacts with the segmenter’s robustness.

5. No reader studies or task-level endpoints. A benchmark can be accepted without reader studies, but then it must convincingly demonstrate practical efficiency (GPU memory/time) and robustness (cross-site, sequence, scanner) advantages over standard pipelines. That evidence is currently thin.

**Questions:**

1. Are encoders frozen with linear probes as the primary metric? If not, can you add this and match head size/epochs across methods, with CIs/paired tests?

2. Do VAEs help when labels are scarce (1–10%)?

3. How do you define and measure “efficiency”? Could you report wall-clock training time, peak VRAM, throughput, encode/decode latency et al.?

4. What is the correlation between PSNR/SSIM and Pillar-2/3 outcomes? If weak, how should practitioners interpret Pillar-1 scores clinically?

5. Why are diffusion/flow autoencoders or latent-diffusion tokenizers excluded, and how might their inclusion change conclusions?

6. Based on results, when should a practitioner choose a VAE over non-generative encoders?

---

### Meta-Review · Area_Chair_qwaR · 2026-01-06

**Summary:**

In the initial phase, all reviewers gave negative scores (2, 4, 2, 4). They raised concerns from different aspects.

Reviewer HX5J has concerns on the assumption regarding VAEs, underlying motivation, and experimental details.
Reviewer sUXW has concerns on the assumption regarding VAEs, missing framework, narrow focus, lack of clinical impacts and cohort-level studies.
Reviewer scbW has concerns on missing related works, and limited coverage of the datasets.
Reviewer yDvX has concerns on dataset usage transparency, limited model diversity, and limited metric novelty.

**Reviewer Concerns:**

Since no rebuttal is provided, all the concerns remain unsolved.

**Reviewer Scores:**

Since no rebuttal is provided, the reviewers may keep their initial scores.

---

### Decision · Program_Chairs · 2026-01-26

Reject